# Association of Depressive and Somatic Symptoms with Heart Rate Variability in Patients with Traumatic Brain Injury

**DOI:** 10.3390/jcm12010104

**Published:** 2022-12-23

**Authors:** Seung Don Yoo, Eo Jin Park

**Affiliations:** Department of Rehabilitation Medicine, Kyung Hee University Hospital at Gangdong, Seoul 05278, Republic of Korea

**Keywords:** heart rate variability, depression, somatic symptom, traumatic brain injury

## Abstract

Depressive and somatic symptoms are common after traumatic brain injury (TBI). Depression after TBI can relate to worsened cognitive functioning, functional impairment, higher rates of suicide attempts, and larger health care costs. Heart rate variability (HRV) represents the activity of the autonomic nervous system (ANS), which regulates almost all vascular, visceral, and metabolic functions. Several studies show a correlation between HRV, depression, and somatic symptoms in other diseases. However, studies on autonomic dysfunction, depression, and somatic symptoms in TBI patients are lacking. This study investigated the association between reduced ANS function, depression, and somatic symptoms in TBI patients. We retrospectively recruited 136 TBI patients who underwent 24 h ambulatory Holter electrocardiography to measure autonomic dysfunction within 1 month of onset. Patients who used BDI and PHQ-15 to evaluate depressive and somatic symptoms were included. Using Pearson’s correlation analysis and multiple linear regression, the association between HRV parameters and BDI and PHQ-15 was determined. The HRV parameters and BDI and PHQ-15 showed statistical significance. In addition, HRV was shown to be a significantly associated factor of BDI and PHQ-15. HRV was associated with depressive and somatic symptom severity in TBI patients. Additionally, autonomic dysfunction may serve as an associated factor of depressive and somatic symptoms in patients with TBI.

## 1. Introduction

Depressive and somatic symptoms are common after traumatic brain injury (TBI). Depression after TBI is reported to occur in nearly half of patients [1,2]. Approximately two weeks following injury, early-onset depression symptoms are often found [3]. Co-occurring somatic symptoms and early-onset depression symptoms are common [4]. Depression after TBI is believed to be related to worse cognitive functioning, increased functional impairment, higher rates of suicide attempts, and larger healthcare costs [2,5,6]. Despite the potential severity of implications associated with depression, the majority of individuals with TBI and depression do not get treatment [7,8]. Because of its multifactored etiology, the diagnosis and management of post-traumatic depression are unclear and challenging [8]. Therefore, the diagnosis of post-TBI depression is important for patient prognosis. The Beck Depression Inventory (BDI) has been widely used to assess depressive symptoms [9]. Somatization is the process through which psychological discomfort emerges as physical symptoms, which may occur with or without intrinsic illness [10]. Several studies have investigated the etiological roles of somatic symptoms in delaying the recovery process after TBI [11,12]. The Patient Health Questionnaire-15 (PHQ-15) is commonly used to evaluate somatic symptoms [13]. Of the most frequent DSM-IV somatization disorder somatic symptoms, 14 of the 15 are included in the PHQ-15 questionnaire [13,14].

The autonomic nervous system (ANS) is made up of sympathetic and parasympathetic systems, and the balance between these activities generally determines the actual effects of the autonomic nervous system on other organs [15]. Heart rate variability (HRV) represents the activity of ANS, which regulates almost all visceral, vascular, and metabolic functions [16,17]. HRV anomalies suggest autonomic instability and are associated with a worse prognosis for cardiovascular disease [18,19]. Several studies have reported low HRV as an independent risk factor for cardiovascular mortality [20,21]. Functional outcomes and activities of daily living were related to abnormalities in frequency domain HRV parameters, including LF and HF [17,22]. Moreover, time domain HRV measures such as rMSSD and SDNN are associated with mortality, functional prognosis, and motor impairment in patients with brain lesions [17,23]. Another study investigated the relationship between the change in RR interval and recovery in patients with TBI in the neurosurgery intensive care unit and found that reduced HRV at any point in the acute recovery period was an indicator of poor outcomes [24].

Several studies have shown a correlation between HRV, depression, and somatic symptoms in other diseases [25,26]. However, studies on autonomic dysfunction, depression, and somatic symptoms in TBI patients are lacking. The purpose of this study was to investigate the association between reduced ANS function, depression, and somatic symptoms in TBI patients.

## 2. Methods

### 2.1. Subjects

A retrospective study was conducted from September 2020 to October 2022 on TBI patients admitted at the Kyung Hee University Hospital in Gangdong, who were diagnosed using computed tomography. Patients who underwent 24 h ambulatory Holter electrocardiography to measure autonomic dysfunction within 1 month of onset and who used BDI and PHQ-15 to evaluate depressive and somatic symptoms were included. To exclude brain lesions that may affect the test results, patients with prior TBI, stroke, or brain tumor were excluded. In addition, patients diagnosed with psychiatric disorders such as depression, somatization disorder, and anxiety disorders before the onset of TBI, and those taking psychiatric medications were excluded. Patients with post-traumatic cranial nerve injury were also excluded. The difference in HRV parameters between groups was evaluated by dividing the groups according to depressive and somatic symptom severity. According to the BDI score severity, groups were divided into a score of 9 or less: normal group (group A); 10 to 18: mild depression group (group B); 19 to 29: moderate depression group (group C); and 30 to 63: severe depression group (group D) [27]. According to the PHQ-15 score, the groups were divided by severity as follows: scores of 4 or less are the normal group (group E), scores of 5 to 9 are the mild somatic symptom group (group F), scores of 10 to 14 are the moderate somatic symptom group (group G), and scores greater than 15 are the severe somatic symptom group (group H) [28]. The study was done according to protocol authorized by the Institutional Review Board (IRB) at Kyung Hee University Hospital in Gangdong, Korea (IRB approval number: 2022-11-021).

### 2.2. Beck Depression Inventory-II

The BDI is a multiple-choice questionnaire consisting of a total of 21 questions [29]. Under each question are four statements, and the participants were told to choose the one that best matched their condition during the previous two weeks. Scores of 0, 1, 2, and 3 were assigned to each question, with 0 being the normal or least depressing statement, and 3 being the most depressing statement. The scores for each item were summed to calculate the total BDI score.

### 2.3. The Patient Health Questionnaire-15

PHQ-15 contains 15 somatic symptoms that account for more than 90% of somatic symptoms [13]. Each item is as follows: (1) stomach pain; (2) back pain; (3) pain in your arms, legs, or joints; (4) menstrual cramps or other problems with your period (women only); (5) headaches; (6) chest pain; (7) dizziness; (8) fainting spells; (9) feeling your heart pound or race; (10) shortness of breath; (11) pain or problems during sexual intercourse; (12) constipation, loose bowels, or diarrhea; (13) nausea, gas, or indigestion; (14) feeling tired or having low energy; and (15) trouble sleeping. In the 13 items of the PHQ somatic symptom module among PHQ-15, the severity of each symptom is checked as 0 points if not bothered at all, 1 point if bothered a little, and 2 points if bothered a lot. The two questions for the PHQ depression module are scored 0 for not at all, 1 for several days, and 2 for more than half the days or nearly every day. To calculate the PHQ-15 score, each symptom is classified as 0, 1, or 2, and the overall score ranges from 0 to 30 [13].

### 2.4. Heart Rate Variability

Three channels of 24 h ambulatory Holter electrocardiography (GE Healthcare, Milwaukee, WI, USA) were used to investigate the HRV. Before the recording, each patient was placed in a supine position for at least 10 minutes in a quiet atmosphere. At a sampling rate of 128 Hz, the digitized data were captured. The R-R interval, the interval between successive heartbeats, and the QRS complex were all measured using the same electrocardiography equipment by a single examiner. Time domain parameters were as follows: root mean square of the difference of successive R-R intervals (rMSSD), standard deviation of the 5 min mean R-R interval (SDANN), standard deviation of the intervals of all normal beat (SDNN), and the mean of 5 min standard deviations of intervals (ASDNN). The frequency domain parameters were as follows: very low frequency (VLF; 0.003–0.04 Hz), HF (0.15–0.40 Hz), LF (0.04–0.15 Hz), and the LF/HF ratio utilizing standard fast Fourier transformation.

### 2.5. Statistical Analysis

The variables were statistically analyzed using SPSS version 20.0 for Windows (IBM Corp., Armonk, NY, USA). The Kolmogorov–Smirnov test was conducted to evaluate the distributional normality of the data. The Levene test was given to examine the variance homogeneity. A one-way analysis of variance (ANOVA) with Bonferroni’s post hoc test was used to compare the HRV parameters across groups. Using Pearson’s correlation coefficient, the relationship between HRV parameters and BDI and PHQ-15 was analyzed. The influence of the HRV parameters on BDI and PHQ-15 was determined using multiple linear regression analysis with stepwise adjustment for sociodemographic factors, lifestyle factors, comorbidities, MMSE, and MBI. In all of the statistical tests, a *p*-value less than 0.05 was regarded as statistically significant.

## 3. Results

### 3.1. Baseline Characteristics

A total of 136 patients were recruited and the mean age was 60.6 ± 12.46 years. Gender consisted of 66 males and 70 females. The mean Mini-Mental State Examination (MMSE) was 23.43 ± 3.32, the modified Barthel index (MBI) was 39.18 ± 18.13, the Beck Depression Inventory (BDI) was 25.59 ± 14.79, and the Patient Health Questionnaire (PHQ) was 14.79 ± 9.08. The baseline characteristics, comorbidities, medication, and HRV parameters are shown in Table 1. Using the Kolmogorov–Smirnov test, all variables were normally distributed (Table 2).

### 3.2. Comparison of the HRV Parameters between Subgroups Classified by Depression Severity

The HRV parameter values of each group are shown in Table 3. In the frequency domain of the HRV parameters, a significant difference in VLF was observed between groups B and D (*p* = 0.001). A significant difference in LF was observed between groups B and D (*p* = 0.018). A significant difference in HF was observed between groups A and D (*p* < 0.001) and B and D (*p* = 0.002). A significant difference in the LF/HF ratio was observed between groups A and B (*p* = 0.026), B and C (*p* = 0.026), and B and D (*p* = 0.007).

In the time domain of the HRV parameters, there was no statistically significant difference in SDNN, SDANN, ASDNN, and pNN50 between groups. A significant difference in rMSSD was observed between groups A and B (*p* = 0.009), B and D (*p* < 0.001), and C and D (*p* = 0.036).

### 3.3. Comparison of the HRV Parameters between Subgroups Classified by Somatic Symptoms Severity

The HRV parameter values of each group are shown in Table 4. In the frequency domain of HRV parameters, a significant difference in VLF was observed between groups F and G (*p* < 0.001) and F and H (*p* < 0.001). A significant difference in LF was observed between groups G and H (*p* = 0.049). A significant difference in HF was observed between groups E and H (*p* = 0.002) and F and H (<0.001). A significant difference in the LF/HF ratio was observed between groups E and F (*p* = 0.015) and F and H (*p* < 0.001).

In the time domain of HRV parameters, there was no statistically significant difference in SDNN, SDANN, ASDNN, and pNN50 between groups. A significant difference in rMSSD was observed between groups F and H (*p* = 0.045) and G and H (*p* = 0.042).

### 3.4. Correlation between HRV Parameters and Depression and Somatic Symptoms Severity

The results of the correlation analysis between the HRV parameters and BDI are shown in Table 5 and Figure 1. In the frequency domains of the HRV parameters, there was a statistically significant correlation between BDI and VLF (r = −0.267, *p* = 0.002), LF (r = −0.192, *p* = 0.025), and HF (r = −0.396, *p* < 0.001). There was no statistically significant correlation between the BDI and LF/HF ratio. In the time domain, BDI and rMSSD (r = −0.193, *p* = 0.025) showed a statistically significant correlation. There was no statistically significant correlation in SDNN, SDANN, ASDNN, and pNN50.

The results of the correlation analysis between the HRV parameters and PHQ-15 are shown in Table 6 and Figure 2. In the frequency domains of the HRV parameters, there was a statistically significant correlation between PHQ-15 and VLF (r = −0.258, *p* = 0.002), LF (r = −0.210, *p* = 0.014), and HF (r = −0.395, *p* < 0.001). There was no statistically significant correlation between the PHQ-15 and LF/HF ratio. In the time domain, PHQ-15 and rMSSD (r = −0.206, *p* = 0.016) showed a statistically significant correlation. There was no statistically significant correlation in SDNN, SDANN, ASDNN, and pNN50.

The results of the multiple linear regression analysis are shown in Table 7. Using the stepwise method, the HF, VLF, and LF variables were finally fitted. However, other variables comprising the LF/HF ratio, SDNN, SDANN, ASDNN, rMSSD, pNN50, sociodemographic factors, lifestyle factors, comorbidities, MMSE, and MBI were excluded from the analysis. HF (standardized β = −0.364, B = −0.080, *p* < 0.001), VLF (standardized β = −0.229, B = −0.016, *p* = 0.003), and LF (standardized β = −0.155, B = −0.015, *p* = 0.045) appeared as significantly associated factors of BDI (adjusted R^2^ = 0.217). HF (standardized β = −0.362, B = −0.049, *p* < 0.001), VLF (standardized β = −0.219, B = −0.009, *p* = 0.005), and LF (standardized β = −174, B = −0.010, *p* = 0.025) appeared as significantly associated factors of PHQ-15 (adjusted R^2^ = 0.219).

## 4. Discussion

To the best of our knowledge, this is the first study to assess the association between the BDI, PHQ-15, and HRV parameters suggesting ANS function in TBI patients. In this study, except for the group with normal depression severity, greater severity was correlated with lower HRV, with a statistically significant difference found between some groups. In the frequency domain and rMSSD, significant differences were observed between the mild and severe depression groups. Similarly, excluding the group with normal somatic symptom severity, greater severity was correlated with lower HRV, with statistically significant differences between some groups. HRV parameters, especially the frequency domain, showed correlations with BDI and PHQ-15. These findings suggest that these HRV parameters are clinically significant for predicting depressive and somatic symptoms.

Acute severe brain injury decreases all natural cyclic heart rate changes [30]. HRV decreases after an acute injury and may be correlated with the future recovery of neurological function [23,31]. Abnormalities in HRV due to brain damage may not be only attributable to cholinergic system failure, but also to increased intracranial pressure [32]. The hypothalamus, brainstem, and cortical processes have been identified as significant regulators of this system. These may include abnormalities in the superficial excitatory effects on brainstem vagus cardiosuppressive neurons, potentially as a result of a breakdown in the connections between these areas [33]. Unbalanced myocardial neuronal activity has been linked to sudden cardiac death and ventricular arrhythmias, which have been attributed to abnormalities in the autonomic regulation of the heart [34,35]. This results in electrical instability as a result of inhomogeneous depolarization and repolarization [34].

The mechanisms of the effects of ANS dysfunction on somatic symptoms and depressive symptoms have not been fully elucidated. Emotional regulation, which is connected with the risk of depression and somatic symptoms, has been linked to the prefrontal cortex and the amygdala, which control cardiac ANS and vagal modulation [36,37]. Consequently, there may be a common pathophysiology involving abnormal ANS function that connects sensitivity to depression, physical symptoms, and cardiac ANS control. Psychological stimuli impact and are influenced by vagal function, as measured by HRV, in ways that might alter long-term morbidity and mortality risks [38]. Reduced vagal activity regulation has been linked to poor physiological systems [36,39]. Vagal stimulation has the effect of improving a depressed mood [40,41]. In our analysis, the strongest association between depression and somatic symptoms was for HF. HF is substantially regulated by the parasympathetic nervous system, indicating that dysregulation of the parasympathetic nervous system, such as a higher vagal withdrawal or a reduced ability for parasympathetic suppression of ANS arousal, is important in depression’s pathogenesis [42]. Patients with somatic symptoms tend to have less parasympathetic activation during emotional tasks [43]. The balanced sympathetic and parasympathetic tone is related to psychological and physiological health [44]. A low sympathetic or parasympathetic activity may contribute to both an autonomic imbalance and the subsequent development of somatic symptoms [44].

Depressive and somatic symptoms influence the functional outcome and prognosis of TBI patients [45]. Therefore, monitoring and managing these symptoms is important for TBI patient rehabilitation. According to our study, HRV parameters were found to be significantly associated factors of depressive and somatic symptoms in TBI patients. These associations suggest that measuring HRV parameters may help monitor depressive and somatic symptoms in patients with TBI.

This study has several limitations. First, it is a retrospective cross-sectional study with a small sample size. Second, single-race subjects were recruited from a single center. Finally, other scales evaluating depressive and somatic symptoms such as the Hamilton Depression Rating Scale, Patient Health Questionnaire-9, Center for Epidemiologic Studies Depression Scale, and Somatic Symptom Scale-8 [46,47,48] were not used in this study. A large-scale prospective longitudinal study using other scales is warranted in the future.

In conclusion, HRV was associated with depressive and somatic symptom severity in TBI patients. In addition, autonomic dysfunction may serve as an associated factor of depressive and somatic symptoms in patients with TBI. The results of this study may serve as an objective basis for evaluating depressive and somatic symptoms that may be overlooked in patients with TBI, helping initiate treatment in due course.

## Figures and Tables

**Figure 1 jcm-12-00104-f001:**
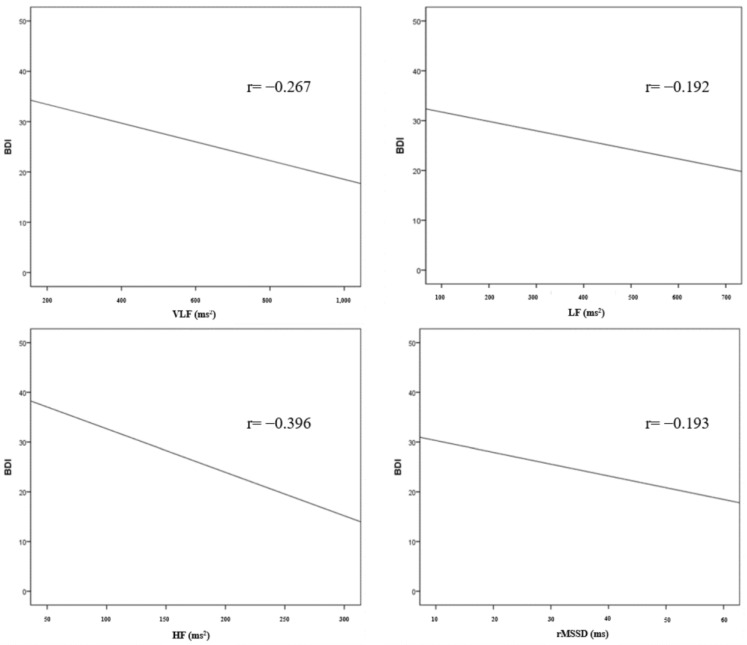
Correlation between HRV parameters and BDI.

**Figure 2 jcm-12-00104-f002:**
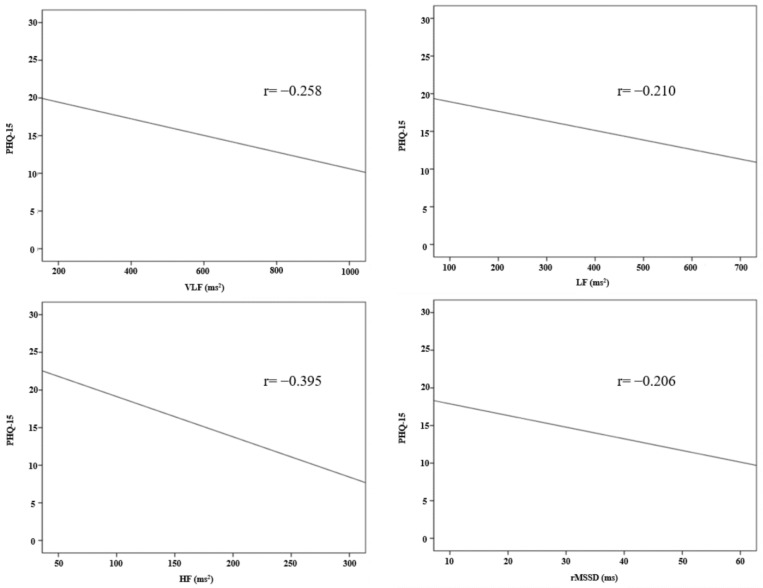
Correlation between HRV parameters and PHQ-15.

**Table 1 jcm-12-00104-t001:** Clinical manifestation of patients.

Characteristic	Value
Sociodemographics	
Age (years)	60.60 ± 12.46
Male	66 (48.50)
Female	70 (51.50)
Lifestyle	
BMI (kg/m^2^)	21.66 ± 2.06
Smoking	278 (65.30)
Regular alcohol use	148 (34.70)
Comorbidities	
Hypertension	83 (61.00)
Arrhythmia	18 (13.20)
Diabetes mellitus	74 (54.40)
Dyslipidemia	67 (49.30)
Coronary artery disease	7 (5.10)
Heart failure	29 (21.30)
Medication	
Beta blocker	33 (24.30)
Calcium channel blocker	87 (64.00)
ACE-i/ARB	82 (60.30)
Diuretics	67 (49.30)
MMSE	23.43 ± 3.32
MBI	39.18 ± 18.13
HRV parameter (frequency domain)	
VLF (ms^2^)	584.62 ± 222.04
LF (ms^2^)	432.46 ± 159.42
HF (ms^2^)	168.54 ± 71.57
LF/HF ratio	5.34 ± 2.75
HRV parameter (time domain)	
SDNN (ms)	98.90 ± 29.57
SDANN (ms)	98.20 ± 29.63
ASDNN (ms)	49.38 ± 11.12
rMSSD (ms)	26.21 ± 6.32
pNN50 (%)	45.06 ± 24.33
Glasgow Coma Scale	
13–15	134 (98.53)
9–12	2 (1.47)
3–8	0 (0.00)
Post-traumatic amnesia	
<24 h	56 (41.18)
1–7 days	3 (2.20)
>7 days	0 (0.00)
No post-traumatic amnesia	77 (56.62)
Loss of consciousness	
<30 min	47 (34.56)
30 min–24 h	5 (3.68)
>24 h	0 (0.00)
No loss of consciousness	84 (61.76)
Cause of injury	
Falls	75 (55.15)
Traffic accidents	38 (27.94)
Violence	16 (11.76)
Sports	7 (5.15)
Skull fracture at injury	32 (23.53)
BDI	25.59 ± 14.79
PHQ-15	14.79 ± 9.08

Values are presented as the mean ± standard deviation or number (%). BMI, Body Mass Index; ACE-I, angiotensin-converting enzyme inhibitors; ARB, angiotensin II receptor blockers; MMSE, Mini-Mental State Examination; MBI, modified Barthel index; HRV, Heart rate variability; VLF, Very low frequency; LF, Low frequency; HF, High frequency; SDNN, standard deviation of intervals of all normal beat; SDANN, standard deviation of 5 min mean R-R interval; ASDNN, mean of 5 min standard deviations of intervals; rMSSD, root mean square of the difference of successive R-R intervals; pNN50, percentage of intervals that are more than 50 ms different from the previous interval; BDI, Beck Depression Inventory; PHQ, Patient Health Questionnaire.

**Table 2 jcm-12-00104-t002:** Kolmogorov–Smirnov test results.

	Statistic	*p*-Value
BDI	0.075	0.051
PHQ-15	0.064	0.200
VLF	0.076	0.050
LF	0.074	0.052
HF	0.066	0.200
LF/HF ratio	0.058	0.200
SDNN	0.072	0.060
SDANN	0.054	0.200
ASDNN	0.059	0.200
rMSSD	0.071	0.061
pNN50	0.050	0.200

BDI, Beck Depression Inventory; PHQ, Patient Health Questionnaire; VLF, very low frequency; LF, low frequency; HF, high frequency; SDNN, standard deviation of intervals of all normal beat; SDANN, standard deviation of 5 min mean R-R interval; ASDNN, mean of 5 min standard deviations of intervals; rMSSD, root mean square of the difference of successive R-R intervals; pNN50, percentage of intervals that are more than 50 ms different from the previous interval.

**Table 3 jcm-12-00104-t003:** Comparison of the HRV parameters between subgroups classified by depression severity.

	Group A (n = 28)	Group B (n = 19)	Group C (n = 32)	Group D (n = 57)	F	*p*-Value	Post Hoc
Frequency domain							
VLF (ms^2^)	668.60	762.92	619.59	550.36	6.002	0.001 *	B > D
	±230.58	±30.15	±139.29	±242.85			
LF (ms^2^)	419.46	505.10	448.73	389.57	3.276	0.023 *	B > D
	±164.01	±142.19	±106.64	±158.56			
HF (ms^2^)	214.75	215.82	176.58	155.27	8.005	<0.001 **	A, B > D
	±57.96	±14.50	±69.57	±69.16			
LF/HF ratio	5.29	7.33	5.34	5.24	3.961	0.010 *	A > B > C, D
	±2.42	±0.52	±2.43	±2.74			
Time domain							
SDNN (ms)	103.08	109.59	104.17	98.60	0.806	0.493	
	±26.60	±27.01	±26.99	±29.86			
SDANN (ms)	106.38	111.13	92.78	97.09	2.606	0.054	
	±29.06	±19.37	±23.69	±29.38			
ASDNN (ms)	49.41	50.70	47.08	48.21	0.611	0.609	
	±11.04	±8.01	±9.40	±10.38			
rMSSD (ms)	27.71	38.57	33.10	26.26	6.863	<0.001 **	A > B, C > D
	±7.65	±16.83	±14.16	±8.49			
pNN50 (%)	49.13	55.65	54.27	47.12	1.229	0.302	
	±20.81	±13.49	±21.90	±23.00			

Values are presented as the mean ± standard deviation. HRV, heart rate variability; VLF, very low frequency; LF, low frequency; HF, high frequency; SDNN, standard deviation of intervals of all normal beat; SDANN, standard deviation of 5 min mean R-R interval; ASDNN, mean of 5 min standard deviations of intervals; rMSSD, root mean square of the difference of successive R-R intervals; pNN50, percentage of intervals that are more than 50 ms different from the previous interval. * *p* < 0.05, ** *p* < 0.001.

**Table 4 jcm-12-00104-t004:** Comparison of the HRV parameters between subgroups classified by somatic symptom severity.

	Group E (n = 28)	Group F (n = 19)	Group G (n = 32)	Group H (n = 57)	F	*p*-Value	Post Hoc
Frequency domain							
VLF (ms^2^)	650.71	763.87	609.11	564.34	5.600	0.001 *	F > G, H
	±241.70	±76.49	±114.96	±235.47			
LF (ms^2^)	431.62	471.19	464.39	390.55	2.796	0.043 *	G > H
	±165.00	±161.30	±118.62	±146.29			
HF (ms^2^)	214.90	217.40	196.04	154.19	9.747	<0.001 **	E, F > H
	±61.93	±21.34	±68.92	±65.72			
LF/HF ratio	5.15	7.06	5.80	5.19	3.659	0.014 *	E > F > H
	±2.49	±1.21	±2.67	±2.56			
Time domain							
SDNN (ms)	102.37	108.05	109.57	97.31	1.559	0.203	
	±28.94	±25.44	±26.09	±28.56			
SDANN (ms)	105.97	110.41	97.44	92.02	2.328	0.078	
	±26.60	±25.81	±23.46	±28.64			
ASDNN (ms)	50.35	48.94	47.71	46.31	0.324	0.808	
	±11.18	±8.76	±9.90	±9.95			
rMSSD (ms)	27.20	35.05	35.00	27.48	4.271	0.007 *	F, G > H
	±7.64	±15.19	±16.13	±9.47			
pNN50 (%)	47.11	57.25	56.31	47.05	2.141	0.098	
	±21.84	±13.77	±22.01	±22.34			

Values are presented as the mean ± standard deviation. HRV, Heart rate variability; VLF, Very low frequency; LF, Low frequency; HF, High frequency; SDNN, standard deviation of intervals of all normal beat; SDANN, standard deviation of 5 min mean R-R interval; ASDNN, mean of 5 min standard deviations of intervals; rMSSD, root mean square of the difference of successive R-R intervals; pNN50, percentage of intervals that are more than 50 ms different from the previous interval. * *p* < 0.05, ** *p* < 0.001.

**Table 5 jcm-12-00104-t005:** Correlation between HRV parameters and the Beck Depression Inventory.

	Pearson Correlation Coefficient (r)	*p*-Value
Frequency domain		
VLF (ms^2^)	−0.267	0.002 *
LF (ms^2^)	−0.192	0.025 *
HF (ms^2^)	−0.396	<0.001 **
LF/HF ratio	−0.086	0.316
Time domain		
SDNN (ms)	−0.077	0.370
SDANN (ms)	−0.128	0.137
ASDNN (ms)	−0.043	0.618
rMSSD (ms)	−0.193	0.025 *
pNN50 (%)	−0.092	0.285

HRV, heart rate variability; VLF, very low frequency; LF, low frequency; HF, high frequency; SDNN, standard deviation of intervals of all normal beat; SDANN, standard deviation of 5 min mean R-R interval; ASDNN, mean of 5 min standard deviations of intervals; rMSSD, root mean square of the difference of successive R-R intervals; pNN50, percentage of intervals that are more than 50 ms different from the previous interval. * *p* < 0.05, ** *p* < 0.001.

**Table 6 jcm-12-00104-t006:** Correlation between HRV parameters and Patient Health Questionnaire-15.

	Pearson Correlation Coefficient (r)	*p*-Value
Frequency domain		
VLF (ms^2^)	−0.258	0.002 *
LF (ms^2^)	−0.210	0.014 *
HF (ms^2^)	−0.395	<0.001 **
LF/HF ratio	−0.091	0.294
Time domain		
SDNN (ms)	−0.084	0.331
SDANN (ms)	−0.106	0.221
ASDNN (ms)	−0.033	0.706
rMSSD (ms)	−0.206	0.016 *
pNN50 (%)	−0.107	0.214

HRV, heart rate variability; VLF, very low frequency; LF, low frequency; HF, high frequency; SDNN, standard deviation of intervals of all normal beat; SDANN, standard deviation of 5 min mean R-R interval; ASDNN, mean of 5 min standard deviations of intervals; rMSSD, root mean square of the difference of successive R-R intervals; pNN50, percentage of intervals that are more than 50 ms different from the previous interval. * *p* < 0.05, ** *p* < 0.001.

**Table 7 jcm-12-00104-t007:** Multiple linear regression analysis HRV parameters as an associated factor of depression and somatic symptom severity.

Dependent Variable	Independent Variable	Standardized *β*	B	95% CI	*p*-Value	VIF	Adjusted *R*^2^
BDI	Constant		56.505				0.217
	HF	−0.364	−0.080	(−0.114, −0.047)	<0.001 **	1.014	
	VLF	−0.229	−0.016	(−0.027, −0.005)	0.003 *	1.009	
	LF	−0.155	−0.015	(−0.030, −0.001)	0.045 *	1.007	
PHQ-15	Constant		33.962				0.219
	HF	−0.362	−0.049	(−0.070. −0.029)	<0.001 **	1.014	
	VLF	−0.219	−0.009	(−0.016, −0.003)	0.005 *	1.009	
	LF	−0.174	−0.010	(−0.020, −0.001)	0.025 *	1.007	

Variables are based on their order of listing in multiple regression analysis. HRV, heart rate variability; B, regression coefficient; CI, confidence interval; VIF, variance inflation factor; BDI, Beck Depression Inventory; PHQ, Patient Health Questionnaire; HF, high frequency; VLF, very low frequency; LF, low frequency. * *p* < 0.05, ** *p* < 0.001.

## Data Availability

The datasets generated and/or analyzed during the current study are available from the corresponding author upon reasonable request.

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
