# Peer review of "Association of Depressive and Somatic Symptoms with Heart Rate Variability in Patients with Traumatic Brain Injury"

_jcm, 2022, doi:10.3390/jcm12010104_

Round 1

Reviewer 1 Report

Dear editors, first of all, thank you for the opportunity to review this text. I attach a number of considerations that I consider to be of interest.

1) Please use more up-to-date references to analyze the current state of the problem. In the paper you cited papers up to 2017 mostly 

2) Please tell more about the limitation of your study 

3) In conclusion please tell more about the future  prospects and practical application of your research

Have a good luck!

Reviewer 2 Report

Thanks for the opportunity to review this paper. While an very interesting topic, the manuscript has to be revised regarding the following comments:

- Information about the TBI definition and characteristics of included patients should be presented such as the injury mechanism, injury severity, extra-cranial injuries.  

- Gender of included patients were 66 males and 70 females. This is somewhat surprising as approximately 70% of TBIs are males. Could you comment on this discrepancy?

- The depressive and somatic symptoms and HRV were assessed at the same point of time so only associations could be presented. However, you have used the predictors term in several places, and this should be replaced with associated factors.

- In line 135 you mentioned electromyography equipment. I recon this is an error – replace with electrocardiography?

- There are a number of statistical analyses performed; thus, the p-values should be adjusted by using for example Bonferroni correction.

- In the multivariate models 21.7% of variance in BDI score is explained by HRV and 21.9% of variance in PHQ-15 score; thus, there is a large amount of variance, which are unexplained. This should be discussed.

- There are five tables with many numbers in this article. The results from table 5 could be more illustrative if presented in the figure.  
